# The relationship between patient experience and real-world digital health access in primary care: A population-based cross-sectional study

Zain Pasat[1,2]*, Chi-Ling Joanna Sinn[1,2☉], Bahram Rahman[1☉], Anastasia Gayowsky[3], Cynthia Lokker[1‡], Jean-Eric Tarride[1,4‡], Mohamed Alarakhia[5,6], Andrew P. Costa[1,2]

1 Department of Health Research Methods, Evidence and Impact, McMaster University, Hamilton, Ontario, Canada, 2 Centre for Integrated Care, St. Joseph's Health System, Hamilton, Ontario, Canada, 3 ICES McMaster, McMaster University Faculty of Health Sciences, Hamilton, Ontario, Canada, 4 Programs for Assessment of Technology in Health (PATH), The Research Institute of St. Joe's Hamilton, St. Joseph's Healthcare, Hamilton, Ontario, Canada, 5 Michael G. DeGroote School of Medicine, McMaster University, Waterloo Regional Campus, Kitchener, Ontario, Canada, 6 eHealth Centre of Excellence, Kitchener, Ontario, Canada

☉ These authors contributed equally to this work.
‡ CL and JET also contributed equally to this work.
* pasatz@mcmaster.ca

**Data Availability Statement:** The dataset from this study is held securely in coded form at ICES. While legal data sharing agreements between ICES and

## Abstract

Implementing digital health technologies in primary care is anticipated to improve patient experience. We examined the relationships between patient experience and digital health access in primary care settings in Ontario, Canada. We conducted a retrospective cross-sectional study using patient responses to the Health Care Experience Survey linked to health and administrative data between April 2019—February 2020. We measured patient experience by summarizing HCES questions. We used multivariable logistic regression stratified by the number of primary care visits to investigate associations between patient experience with digital health access and moderating variables. Our cohort included 2,692 Ontario adults, of which 63.0% accessed telehealth, 2.6% viewed medical records online, and 3.6% booked appointments online. Although patients reported overwhelmingly positive experiences, we found no consistent relationship with digital health access. Online appointment booking access was associated with lower odds of poor experience for patients with three or more primary care visits in the past 12 months (adjusted odds ratio 0.16, 95% CI 0.02–0.56). Younger age, tight financial circumstances, English as a second language, and knowing their primary care provider for fewer years had greater odds of poor patient experience. In 2019/2020, we found limited uptake of digital health in primary care and no clear association between real-world digital health adoption and patient experience in Ontario. Our findings provide an essential context for ensuing rapid shifts in digital health adoption during the COVID-19 pandemic, serving as a baseline to reexamine subsequent improvements in patient experience.

data providers (e.g., healthcare organizations and government) prohibit ICES from making the dataset publicly available, access may be granted to those who meet prespecified criteria for confidential access, available at https://www.ices. on.ca/DAS/AHRQ (email: das@ices.on.ca). The full dataset creation plan and underlying analytic code have been included as supplemental information files.

**Funding:** The work was originally a thesis internally funded by operating funds from St. Joseph's Centre for Integrated Care. There is no grant funding number available.

**Competing interests:** The authors have declared that no competing interests exist.

## Introduction

Patient-centred, personalized care is essential in establishing quality care [1–6]. Patient experience measures frequently operationalize centeredness, allowing us to evaluate how patients receive care consistent with their goals [4–7]. Patient experience encompasses all patient interactions influencing perceptions of a health system across the continuum of care, including shared decision-making, coordinated care, information sharing, and improved access [6, 8–10].

Digital health technologies can enhance patient experience and patient-provider communication by supporting accessible and timely information flow and team-based processes [11–14]. Over the past few decades, organizations have spearheaded Ontario's digital health adoption within primary care to improve patient-centeredness and provider-to-provider care coordination. By 2019, 85% of family physicians had adopted certified electronic medical record (EMR) systems, while over 600,000 Ontario Telemedicine Network (OTN) visits were completed between 2008/2009 and 2013/2014 [1, 15–21]. Despite these developments, Ontario's efforts to digitize primary care were fragmented, and adoption rates plateaued before the COVID-19 pandemic, with virtual visits accounting for only 1.2% of primary care visits in 2019 [22, 23].

There is a growing need to examine whether the experimental evidence for digital health translates to the real-world context of individuals with wide-ranging health and digital literacies using various technologies. Past evaluations of digital health tools were efficacy-based and did not provide evidence for the everyday use of technologies or diverse groups of users [24–26]. For example, a cohort study of Ontario primary care patients and providers demonstrated that virtual care improved access and patient experience [24]. However, the intervention was limited to synchronous and asynchronous virtual care [24].

Our objective was to examine if pre-pandemic access to telehealth, electronic health records (EHRs), and online appointment booking are associated with the primary care experience of Ontario adults. Our findings describe the influence of digital health and predictors on patient experience and act as a baseline for repeated studies examining rapid digital health adoption and patient experience beyond the pandemic. We hypothesize that digital health access is associated with favourable primary care experiences.

## Methods

### Study design and participants

We conducted a cross-sectional retrospective study to evaluate the association between primary care experience and digital health access in Ontarians ≥16 years of age. We derived the cohort from the Health Care Experience Survey (HCES). The HCES is a cross-sectional survey used by the Ontario Ministries of Health and Long-Term Care and researchers to measure access and experience across the healthcare system [27–30]. The Institute for Social Research (ISR) at York University conducts the HCES. The sampling frame of the HCES uses the Registered Persons Database (RPDB) to randomly select households of Ontario adults aged 16 years or older. A notification letter is mailed to households selected for the survey, followed by a phone interview (via cellphone or landline) [31]. Participants may refuse to answer questions on the HCES and may choose to stop the interview at any point. ISR also collects informed consent from participants over the phone interview to link HCES responses with administrative databases within ICES [30]. The response rate for included waves ranged from 29% to 39% [27].

We accessed all de-identified HCES responses linked to administrative datasets through ICES (formally Institute for Clinical Evaluative Sciences) collected between October 28, 2012, and February 20, 2020. ICES fully anonymized all data before being accessed by investigators.

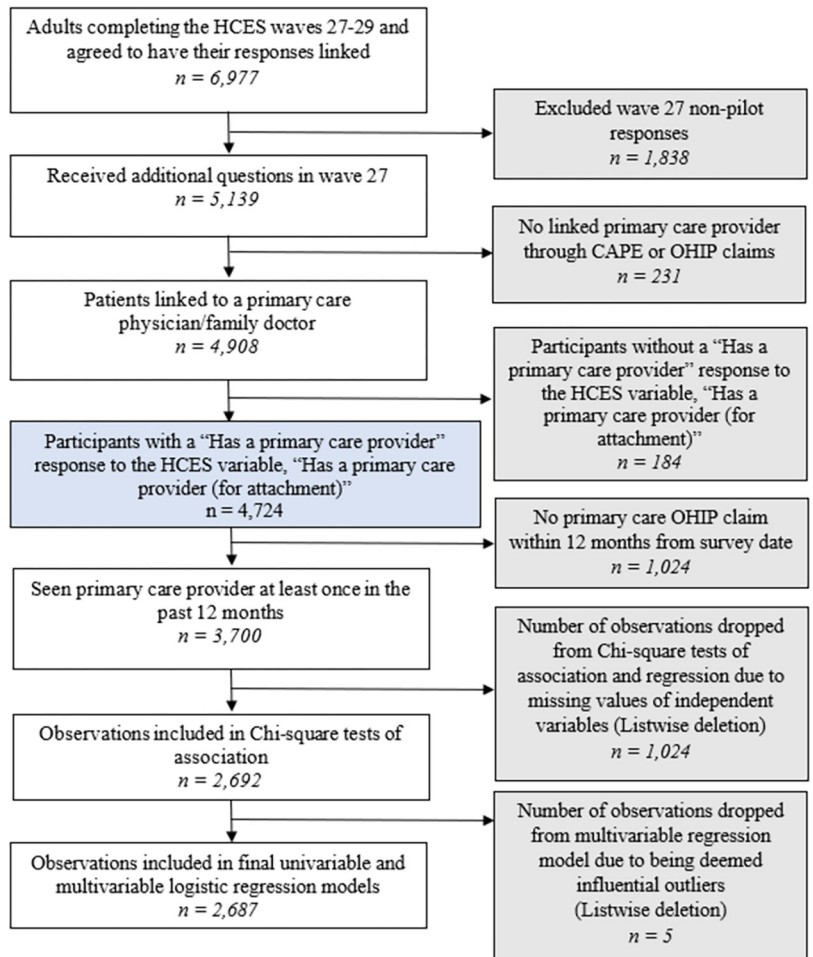

**Fig 1. Flow diagram of survey respondents included in the study cohort.**

Our analyses included participants who responded to the HCES from the wave 27 pilot, which introduced digital health questions, to wave 29, the latest data at the time of analysis, as the survey was paused due to the COVID-19 pandemic (April 1, 2019 –February 20, 2020). Participants agreed to have their responses linked with ICES data holdings. We excluded adults who were not linked to or did not self-report having a primary care provider or having a primary care visit within 12 months of the survey date, as several HCES questions are framed within 12 months (Fig 1).

## Dependent measures

Like prior studies, our patient experience measure was a linear combination of Likert responses (1 to 5, never to always) to HCES questions on patient experience (Table 1) [32, 33]. We removed responses: *It depends [on] who they see/what they are there for*, *Not Applicable*, *Don't know*, and *Refused* from the analysis.

## Digital health access

The HCES includes information on access to telehealth, EHRs, and online appointment booking. We classified telehealth access by seven grouped HCES questions regarding primary care

**Table 1. Coding of summed experience score.**

| Experience questions | Coding |
|---|---|
| When you see your provider or someone else in their office, how often do they know important information about your medical history? | Responses were numerically coded to values from 1 ("never") to 5 ("always") and summed. |
| When you see your provider or someone else in their office, how often do they give you an opportunity to ask questions about recommended treatment? | |
| When you see your provider or someone else in their office, how often do they spend enough time with you? | |
| When you see your provider or someone else in their office, how often do they involve you as much as you want to be in decisions about your care and treatment? | |
| When you see your provider or someone else in their office, how often do they explain things in a way that is easy to understand? | |

access over the past 12 months through telephone, email, video, electronic messaging, or other virtual means. We determined EHR access using three HCES questions that assessed whether participants had viewed medical records using an online system or digital tool over the past 12 months, including systems designed for people with specific health conditions or comprehensive health records. We identified online appointment booking access using one HCES question to which participants responded whether they emailed or visited a website to set up a primary care appointment over the past 12 months.

## Covariables

We included variables in the analysis based on hypothesized associations with patient experience, past literature, and frameworks such as Levesque's conceptual framework for healthcare access [33, 34]. These variables included self-reported financial situation, primary language spoken at home, educational attainment, health status, and years of knowing their primary care provider from the HCES. The RPDB provided the age and sex of participants. We obtained the Rurality Index of Ontario (RIO) from the RPDB and coded it into groups of large urban (RIO: 0), urban (RIO: 1–9), small urban (RIO: 10–39), and rural (RIO: $\geq$ 40) [35]. We obtained Aggregated Diagnoses Groups (ADG) through the Johns Hopkins ACG® System Version 10. We identified primary care providers rostered or virtually rostered to patients by the highest payment of primary care services over the past 12 months through the Client Agency Program Enrolment (CAPE) and the Ontario Health Insurance Plan Claims (OHIP) databases. We grouped patient enrolment models provided by the Corporate Provider Database (CPDB) and the ICES Physician Database (IPDB) into the following categories: Enhanced Fee-for-Service, Capitation, and Other. OHIP provided the number of primary care visits with the most responsible provider 12 months before the survey date. The Ontario Marginalization Index (ONMARG) provided quintiles for dissemination-area concentrations of material deprivation, dependency, ethnic concentration, and residential instability. S1 Appendix describes the coding of independent variables.

## Statistical methods

We compared the characteristics of respondents with missing data to those with complete data to assess bias (systematic vs random missingness) and implemented row-wise deletion for missing covariates. Per communication from the Ministry of Health and the ISR, we coded missing responses to digital health questions as "no access" in line with skip patterns.

We fit an unstratified multivariable logistic regression model with patient experience as the outcome. We included potential confounders through a hierarchical approach, first implementing digital health access, patient demographics and health, healthcare-related factors, and geographic factors [33, 36, 37]. We tested the independence assumption through clustering by providers and LHIN. We fit additional models stratified by the median number of visits.

We conducted a sub-group analysis using multivariable logistic regression to measure how digital health factors were associated with participants reporting "sometimes," "rarely," or "never" to individual HCES experience items, unstratified and stratified by the median age and number of encounters. We removed influential outliers from all models and observed multicollinearity through generalized variance inflation factors (GVIF). We assessed goodness of fit using the c-statistic. A c-statistic larger than 0.70 indicates a good model fit. All analyses were conducted using R Studio (version 1.1.456).

## Ethics approval

The Hamilton Integrated Research Ethics Board exempted this study from a formal ethics review as this research project falls under section 45 of Ontario's Personal Health Information Protection Act, which does not require a research ethics board review. The use of the data in this study is authorized and approved by ICES' Privacy and Legal Office, which does not require participant consent according to regulations in Ontario, Canada.

## Results

### Participants & descriptive statistics

We included 2,692 participants (Fig 1). Table 2 provides our sample demographics. Compared to EHR (2.6%) and online appointment booking access (3.6%), telehealth access was more prominent (63.0%), primarily attributed to telephone-based access. Most participants reported overwhelmingly positive patient experience, with a median score of 24 out of the potential 25 (IQR 22–25) (S2 Appendix). Six hundred twenty-five participants (23.2%) were categorized to have poorer primary care experience (1st quintile and below [score ≤ 21]), and 2067 participants (76.8%) had positive primary care experiences (remaining sample [score > 21]).

### Main results

**Summed patient experience.** We found no significant association between patient experience and telehealth access (p = 0.45), digital health record access (p = 1.00), and online booking access (p = 0.46). Tables 3 and 4 provide odds ratios for unstratified and stratified models by the number of encounters, respectively. Random effects were not considered in the final model, as the mixed-effect models did not improve fit, and each provider was only responsible for one to two participants. Generalized variance inflation factors did not suggest multicollinearity. The unstratified model had a c-statistic of 0.65, which remained consistent after stratification (≤ 3 encounters: 0.61; > three encounters: 0.66).

After adjusting, telehealth (aOR 1.00, 95% CI 0.77–1.28), digital health record (aOR 0.97, 95% CI 0.41–2.10), and online appointment booking access (aOR 1.07, 95% CI 0.57–1.93) were not associated with poor experience in participants with three or fewer encounters. In those with over three visits in the past 12 months, participants with online booking access had an 84% reduction in the odds of reporting poor primary care experience than those with no online booking access over the past 12 months (aOR 0.16, 95% CI 0.02–0.56). However, telehealth access (aOR 0.83, 95% CI 0.59–1.17) and digital health record access (aOR 0.82, 95% CI 0.31–1.92) were nonsignificant in the same subgroup.

**Table 2. Digital health access, personal, health and care-related, and geographic characteristics of the cohort.**

| Variables | Full cohort, N = 3,700 | Removed missing values, N = 2,692 | Summed patient experience score | |
|---|---|---|---|---|
| | | | Poor experience, N = 625 | Positive experience, N = 2,067 |
| *Outcome Measure* | | | | |
| **Summed primary care experience** | | | | |
| Positive experience[1] **[% (n)]** | | 23.2% (2067) | | |
| Poor experience[1] **[% (n)]** | | 76.8% (625) | | |
| Score (/25) [median (IQR)] | | 24 (22–25) | | |
| *Independent Variables* | | | | |
| **Telehealth access[2] [% (n)]** | | | | |
| Yes | 59.6% (2204) | 63.0% (1695) | 61.6% (385) | 63.4% (1310) |
| No | 39.8% (1474) | 37.0% (997) | 38.4% (240) | 36.6% (757) |
| Missing | 0.6% (22) | | | |
| **Health record access[3] [% (n)]** | | | | |
| Yes | 2.6% (98) | 2.6% (71) | 2.6% (16) | 2.7% (55) |
| No | 97.1% (3593) | 97.4% (2621) | 97.4% (609) | 97.3% (2012) |
| Missing | 0.2% (9) | | | |
| **Online booking access [% (n)]** | | | | |
| Yes | 3.2% (120) | 3.6% (97) | 3.0% (19) | 3.8% (78) |
| No | 96.3% (3563) | 96.4% (2595) | 97.0% (606) | 96.2% (1989) |
| Missing | 0.5% (17) | | | |
| **Age (years) [% (n)]** | | | | |
| 16–44 | 22.9% (849) | 23.6% (636) | 35.4% (221) | 20.1% (415) |
| 45–64 | 35.9% (1328) | 36.9% (993) | 36.3% (226) | 37.1% (767) |
| 65+ | 41.2% (1523) | 39.5% (1063) | 28.5% (178) | 42.8% (885) |
| **Sex** | | | | |
| Female | 61.4% (2271) | 60.7% (1633) | 63.4% (396) | 59.8% (1237) |
| Male | 38.6% (1429) | 39.3% (1059) | 36.6% (229) | 40.2% (830) |
| **Financial situation [% (n)]** | | | | |
| Very comfortable | 17.8% (658) | 19.2% (517) | 15.2% (95) | 20.4% (422) |
| Comfortable | 59.2% (2192) | 61.8% (1663) | 59.4% (371) | 62.5% (1292) |
| Tight/Very tight/Poor | 19.4% (719) | 19.0% (512) | 25.4% (159) | 17.1% (353) |
| Missing | 3.5% (131) | | | |
| **Primary language [% (n)]** | | | | |
| English | 86.0% (3182) | 87.4% (2352) | 81.0% (506) | 89.3% (1846) |
| Other | 13.8% (511) | 12.6% (340) | 19.0% (119) | 10.7% (221) |
| Missing | 0.2% (7) | | | |
| **Educational attainment [% (n)]** | | | | |
| High school or less | 27.8% (1029) | 25.6% (690) | 24.5% (153) | 26.0% (537) |
| Some college/university | 8.9% (330) | 8.4% (227) | 8.0% (50) | 8.6% (177) |
| Completed college/university | 49.6% (1834) | 52.4% (1410) | 55.2% (345) | 51.5% (1065) |
| Post-graduate/professional degree | 13.0% (481) | 13.6% (365) | 12.3% (77) | 13.9% (288) |
| Missing | 0.7% (26) | | | |
| **Dependency[4] [% (n)]** | | | | |
| 1st quintile *(least marginalized)* | 20.4% (755) | 20.3% (546) | 23.2% (145) | 19.4% (401) |
| 2nd quintile | 17.9% (662) | 18.5% (499) | 20.2% (126) | 18.0% (373) |
| 3rd quintile | 19.5% (720) | 19.4% (523) | 20.0% (125) | 19.3% (398) |
| 4th quintile | 18.2% (675) | 18.2% (489) | 17.9% (112) | 18.2% (377) |

*(Continued)*

**Table 2.** (Continued)

| Variables | Full cohort, N = 3,700 | Removed missing values, N = 2,692 | Summed patient experience score | |
|---|---|---|---|---|
| | | | Poor experience, N = 625 | Positive experience, N = 2,067 |
| 5th quintile (most marginalized) | 23.4% (867) | 23.6% (635) | 18.7% (117) | 25.1% (518) |
| Missing | 0.6% (21) | | | |
| **Material deprivation[5] [% (n)]** | | | | |
| 1st quintile (least marginalized) | 24.2% (895) | 24.8% (668) | 24.6% (154) | 24.9% (514) |
| 2nd quintile | 22.6% (835) | 23.5% (633) | 23.0% (144) | 23.7% (489) |
| 3rd quintile | 19.1% (706) | 19.5% (524) | 18.9% (118) | 19.6% (406) |
| 4th quintile | 18.2% (672) | 17.4% (469) | 18.7% (117) | 17.0% (352) |
| 5th quintile (most marginalized) | 15.4% (571) | 14.8% (398) | 14.7% (92) | 14.8% (306) |
| Missing | 0.6% (21) | | | |
| **Ethnic concentration[6] [% (n)]** | | | | |
| 1st quintile (least marginalized) | 21.4% (793) | 21.4% (575) | 17.9% (112) | 22.4% (463) |
| 2nd quintile | 18.8% (697) | 19.2% (518) | 17.4% (109) | 19.8% (409) |
| 3rd quintile | 19.4% (716) | 20.2% (545) | 17.3% (108) | 21.1% (437) |
| 4th quintile | 19.3% (714) | 19.9% (535) | 22.2% (139) | 19.2% (396) |
| 5th quintile (most marginalized) | 20.5% (759) | 19.3% (519) | 25.1% (157) | 17.5% (362) |
| Missing | 0.6% (21) | | | |
| **Residential instability[7] [% (n)]** | | | | |
| 1st quintile (least marginalized) | 19.9% (738) | 20.0% (539) | 20.2% (126) | 20.0% (413) |
| 2nd quintile | 19.6% (725) | 19.7% (529) | 20.5% (128) | 19.4% (401) |
| 3rd quintile | 19.4% (719) | 19.8% (532) | 17.1% (107) | 20.6% (425) |
| 4th quintile | 18.6% (690) | 19.1% (514) | 19.8% (124) | 18.9% (390) |
| 5th quintile (most marginalized) | 21.8% (807) | 21.5% (578) | 22.4% (140) | 21.2% (438) |
| Missing | 0.6% (21) | | | |
| **RIO category[8] [% (n)]** | | | | |
| Large urban (RIO: 0) | 39.1% (1446) | 39.5% (1062) | 42.1% (263) | 38.7% (799) |
| Urban (RIO: 1–9) | 26.7% (989) | 26.5% (713) | 27.5% (172) | 26.2% (541) |
| Small urban (RIO: 10–40) | 23.5% (868) | 24.3% (654) | 22.2% (139) | 24.9% (515) |
| Rural (RIO: > 40) | 9.7% (360) | 9.8% (263) | 8.2% (51) | 10.3% (212) |
| Missing | 1.0% (37) | | | |
| **Self-reported health [% (n)]** | | | | |
| Poor | 5.2% (193) | 4.8% (128) | 4.6% (29) | 4.8% (99) |
| Fair | 15.2% (563) | 14.7% (396) | 16.3% (102) | 14.2% (294) |
| Good | 32.0% (1184) | 32.8% (884) | 35.8% (224) | 21.9% (660) |
| Very good | 33.5% (1240) | 34.1% (917) | 32.8% (205) | 34.4% (712) |
| Excellent | 13.5% (498) | 13.6% (367) | 10.4% (65) | 14.6% (302) |
| Missing | 0.6% (22) | | | |
| **ADG Score[9] [% (n)]** | | | | |
| <3 | 9.9% (365) | 9.9% (266) | 9.6% (60) | 10.0% (206) |
| 3–4 | 22.4% (830) | 23.0% (618) | 24.6% (154) | 22.4% (464) |
| 5–6 | 24.1% (892) | 24.0% (646) | 21.4% (134) | 24.8% (512) |
| 7–8 | 20.2% (746) | 20.4% (548) | 19.7% (123) | 20.6% (425) |
| ≥9 | 23.4% (867) | 22.8% (614) | 24.6% (154) | 22.3% (460) |
| **Program type [% (n)]** | | | | |
| Enhanced Fee-for-Service | 24.1% (892) | 26.6 (717) | 32.8% (205) | 24.8% (512) |
| Capitation | 64.2% (2376) | 71.4% (1923) | 65.8% (411) | 73.1% (1512) |

(Continued)

**Table 2.** (Continued)

| Variables | Full cohort, N = 3,700 | Removed missing values, N = 2,692 | Summed patient experience score | |
|---|---|---|---|---|
| | | | Poor experience, N = 625 | Positive experience, N = 2,067 |
| Other | 2.0% (74) | 1.9% (52) | 1.4% (9) | 2.1% (43) |
| Missing | 9.7% (358) | | | |
| **Number of years with provider [% (n)]** | | | | |
| Less than 3 | 22.8% (843) | 22.0% (593) | 27.0% (169) | 20.5% (424) |
| 4–9 | 23.5% (869) | 23.8% (641) | 23.5% (147) | 23.9% (494) |
| 10–19 | 22.5% (834) | 23.3% (626) | 24.2% (151) | 23.0% (475) |
| 20 or more | 29.6% (1096) | 30.9% (832) | 25.3% (158) | 32.6% (674) |
| Missing | 1.6% (58) | | | |
| **Primary care encounters with primary provider over 12 months (median, IQR)** | 3 (2–5) | 3 (2–5) | 3 (2–5) | 3 (2–5) |

[1]Defined as summed first quintile of patient experience scores ($\leq 21$)

[2]Includes telephone, email, video, electronic messaging, or other virtual means of communication with primary care provider 12 months before the survey

[3]Includes digital medical record access 12 months before the survey, including systems specific to health conditions or comprehensive records

[4]Defined by the Ontario Marginalization Index as the proportion of the population without an income generated from employment, including ratios of seniors and unemployment within the population

[5]Defined by the Ontario Marginalization Index as a measure of poverty and access to basic human needs

[6]Defined by the Ontario Marginalization Index as the concentration of recent immigrants and visible minorities

[7]Defined by the Ontario Marginalization Index as the level of family and housing instability

[8]Rurality Index of Ontario

[9]John Hopkins Aggregated Diagnosis Groups

Other factors, including younger age, tight financial circumstances, primary languages other than English, and fewer years with a dedicated primary care provider, were associated with less favourable experiences. Financially and ethnically marginalized communities reported poorer experiences, consistent with financial comfort and language barriers (Tables 3 and 4).

**Sub-group analysis.** Participants who accessed health records electronically had a 65% decrease in the odds of reporting their primary care provider sometimes, rarely, or never being aware of details of their medical history compared to those without access (aOR 0.35, 95% CI 0.11–0.88). In addition, participants with greater than three encounters over the past 12 months who used telehealth had a 37% decrease in odds of reporting their primary care provider sometimes, rarely, or never providing the opportunity to ask questions compared to those without access (aOR 0.63, 95% CI 0.41–0.96). Older adults over 60 who accessed telehealth also had a 33% decrease in the odds of reporting their provider sometimes, rarely, or never spent enough time with them compared to the same reference group (aOR 0.67, 95% CI 0.46–0.98). We observed other associations among experience items and confounders (S3 Appendix).

## Discussion

### Interpretation

We found no consistent association between access to online appointment booking and digital health records and primary care patient experience. However, our sub-group analysis suggested telehealth access was associated with enhanced communication and time spent with

**Table 3. Associations between digital health, personal, healthcare, and geographic factors with poor primary care experience (N = 2,687).**

| Factors | Poor Patient Experience[1] (As of May 2019—February 2020) OR (95% CI) | | | |
|---|---|---|---|---|
| | Odds ratio (OR) | p | Adjusted odds ratio (aOR) | p |
| **Telehealth access[2]** | 0.94 (0.78–1.13) | 0.489 | 0.94 (0.77–1.14) | 0.529 |
| **Health record access[3]** | 0.96 (0.53–1.65) | 0.895 | 0.90 (0.48–1.60) | 0.739 |
| **Online booking access** | 0.80 (0.47–1.30) | 0.393 | 0.69 (0.39–1.16) | 0.181 |
| **Age (years)** | | | | |
| 16–44 | Reference | | Reference | |
| 45–64 | 0.55 (0.44–0.69) | **<0.001** | 0.57 (0.45–0.73) | **<0.001** |
| $\geq$ 65 | 0.38 (0.30–0.48) | **<0.001** | 0.40 (0.31–0.53) | **<0.001** |
| **Sex** | | | | |
| Female | Reference | | Reference | |
| Male | 0.87 (0.72–1.04) | 0.127 | 0.88 (0.72–1.07) | 0.214 |
| **Financial situation** | | | | |
| Very comfortable | Reference | | Reference | |
| Comfortable | 1.27 (0.99–1.64) | 0.063 | 1.07 (0.83–1.40) | 0.616 |
| Tight/Very tight/Poor | 1.99 (1.49–2.68) | **<0.001** | 1.49 (1.09–2.04) | **0.013** |
| **Educational attainment** | | | | |
| High school or less | Reference | | Reference | |
| Some college/university | 1.00 (0.69–1.43) | 0.988 | 0.89 (0.61–1.29) | 0.543 |
| Completed college/university | 1.13 (0.91–1.41) | 0.253 | 0.96 (0.76–1.22) | 0.734 |
| Post-graduate/professional degree | 0.93 (0.68–1.26) | 0.628 | 0.82 (0.58–1.14) | 0.242 |
| **Primary language spoken** | | | | |
| English | Reference | | Reference | |
| Other | 1.95 (1.52–2.48) | **<0.001** | 1.49 (1.12–1.95) | **0.005** |
| **Self-perceived health** | | | | |
| Poor | Reference | | Reference | |
| Fair | 1.19 (0.75–1.93) | 0.473 | 1.23 (0.76–2.02) | 0.415 |
| Good | 1.15 (0.75–1.82) | 0.526 | 1.14 (0.73–1.85) | 0.575 |
| Very good | 0.98 (0.64–1.55) | 0.932 | 0.93 (0.59–1.52) | 0.780 |
| Excellent | 0.73 (0.45–1.22) | 0.220 | 0.65 (0.38–1.11) | 0.110 |
| **ADG Score[4]** | | | | |
| < 3 | Reference | | Reference | |
| 3–4 | 1.13 (0.81–1.60) | 0.468 | 1.13 (0.79–1.61) | 0.508 |
| 5–6 | 0.90 (0.64–1.27) | 0.543 | 0.89 (0.62–1.29) | 0.536 |
| 7–8 | 0.99 (0.70–1.41) | 0.956 | 0.92 (0.64–1.35) | 0.670 |
| $\geq$ 9 | 1.15 (0.82–1.62) | 0.423 | 1.17 (0.81–1.71) | 0.407 |
| **Program type** | | | | |
| Enhanced FFS[5] | Reference | | Reference | |
| Capitation | 0.68 (0.56–0.83) | **<0.001** | 0.85 (0.68–1.07) | 0.156 |
| Other | 0.54 (0.24–1.07) | 0.098 | 0.70 (0.31–1.47) | 0.375 |
| **Years with provider** | | | | |
| < 4 | Reference | | Reference | |
| 4–9 | 0.75 (0.58–0.96) | **0.025** | 0.73 (0.56–0.95) | **0.019** |
| 10–19 | 0.79 (0.61–1.02) | 0.075 | 0.81 (0.62–1.06) | 0.128 |
| $\geq$ 20 | 0.59 (0.46–0.75) | **<0.001** | 0.68 (0.52–0.89) | **0.004** |
| **RIO category[6]** | | | | |
| Large urban | Reference | | Reference | |
| Urban | 0.96 (0.77–1.20) | 0.724 | 0.93 (0.74–1.18) | 0.575 |

*(Continued)*

**Table 3.** (Continued)

| Factors | Poor Patient Experience[1] (As of May 2019—February 2020) OR (95% CI) | | | |
| --- | --- | --- | --- | --- |
| | Odds ratio (OR) | p | Adjusted odds ratio (aOR) | p |
| Small urban | 0.81 (0.64–1.03) | 0.083 | 1.10 (0.81–1.48) | 0.545 |
| Rural | 0.73 (0.52–1.02) | 0.071 | 1.07 (0.70–1.63) | 0.741 |
| **Dependency[7]** | | | | |
| 1st quintile (least marginalized) | Reference | | Reference | |
| 2nd quintile | 0.94 (0.71–1.24) | 0.680 | 1.16 (0.86–1.57) | 0.323 |
| 3rd quintile | 0.87 (0.66–1.15) | 0.343 | 1.14 (0.83–1.55) | 0.420 |
| 4th quintile | 0.82 (0.62–1.09) | 0.178 | 1.13 (0.82–1.57) | 0.452 |
| 5th quintile (most marginalized) | 0.63 (0.48–0.83) | **0.001** | 0.95 (0.68–1.34) | 0.782 |
| **Material Deprivation[8]** | | | | |
| 1st quintile (least marginalized) | Reference | | Reference | |
| 2nd quintile | 0.98 (0.76–1.27) | 0.884 | 0.87 (0.66–1.15) | 0.332 |
| 3rd quintile | 0.97 (0.74–1.27) | 0.830 | 0.85 (0.63–1.14) | 0.284 |
| 4th quintile | 1.10 (0.83–1.45) | 0.496 | 0.92 (0.67–1.26) | 0.591 |
| 5th quintile (most marginalized) | 0.99 (0.74–1.33) | 0.950 | 0.70 (0.49–0.99) | **0.046** |
| **Ethnic Concentration[9]** | | | | |
| 1st quintile (least marginalized) | Reference | | Reference | |
| 2nd quintile | 1.11 (0.82–1.49) | 0.492 | 1.10 (0.80–1.51) | 0.570 |
| 3rd quintile | 1.03 (0.77–1.39) | 0.840 | 1.01 (0.71–1.43) | 0.977 |
| 4th quintile | 1.46 (1.10–1.95) | **0.008** | 1.39 (0.95–2.02) | 0.088 |
| 5th quintile (most marginalized) | 1.79 (1.36–2.38) | **<0.001** | 1.43 (0.95–2.14) | 0.085 |
| **Residential Instability[10]** | | | | |
| 1st quintile (least marginalized) | Reference | | Reference | |
| 2nd quintile | 1.04 (0.78–1.38) | 0.795 | 1.16 (0.86–1.57) | 0.345 |
| 3rd quintile | 0.82 (0.61–1.10) | 0.190 | 1.00 (0.73–1.37) | 0.999 |
| 4th quintile | 1.03 (0.78–1.37) | 0.832 | 1.22 (0.89–1.69) | 0.220 |
| 5th quintile (most marginalized) | 1.05 (0.80–1.38) | 0.741 | 1.18 (0.85–1.63) | 0.314 |
| **C-Statistic** | | | 0.65 | |

[1]Defined as summed first quintile of patient experience scores ($\leq 21$)

[2]Includes telephone, email, video, electronic messaging, or other virtual means of communication with primary care provider 12 months before the survey

[3]Includes digital medical record access 12 months before the survey, including systems specific to health conditions or comprehensive records

[4]John Hopkins Aggregated Diagnosis Groups

[5]Enhanced fee for service

[6]Rurality Index of Ontario

[7]Defined by the Ontario Marginalization Index as the proportion of the population without an income generated from employment, including ratios of seniors and unemployment within the population

[8]Defined by the Ontario Marginalization Index as a measure of poverty and access to basic human needs

[9]Defined by the Ontario Marginalization Index as the concentration of recent immigrants and visible minorities

[10]Defined by the Ontario Marginalization Index as the level of family and housing instability

primary care providers. Patient experience is likely influenced more heavily by other provider factors rather than digital health tools as facilitators of patient-centeredness.

Contrary to our findings, past research has observed that patient access to digital health records improves patient-provider communication and overall experiences [14, 38]. A need for greater adoption to measure any potential effect may explain these mixed outcomes. Digital health technologies are enablers rather than solutions, and providers need payment

**Table 4. Associations between digital health, personal, healthcare, and geographic factors, with poor primary care experience (N = 2,687).**

| Factors | Poor Patient Experience[1] (As of May 2019—February 2020) OR (95% CI) | | | |
|---|---|---|---|---|
| | ≤ 3 encounters, N = 1,647 | p | > 3 encounters, N = 1,038 | p |
| **Telehealth access[2]** | 1.00 (0.77–1.28) | 0.976 | 0.83 (0.59–1.17) | 0.284 |
| **Health record access[3]** | 0.97 (0.41–2.10) | 0.933 | 0.82 (0.31–1.92) | 0.661 |
| **Online booking access** | 1.07 (0.57–1.93) | 0.838 | 0.16 (0.02–0.56) | **0.015** |
| **Age (years)** | | | | |
| 16–44 | Reference | | Reference | |
| 45–64 | 0.59 (0.44–0.80) | **0.001** | 0.49 (0.31–0.75) | **0.001** |
| ≥ 65 | 0.44 (0.31–0.62) | **<0.001** | 0.32 (0.20–0.51) | **<0.001** |
| **Sex** | | | | |
| Female | Reference | | Reference | |
| Male | 0.87 (0.68–1.12) | 0.289 | 0.91 (0.64–1.27) | 0.564 |
| **Financial situation** | | | | |
| Very comfortable | Reference | | Reference | |
| Comfortable | 1.06 (0.76–1.49) | 0.749 | 1.08 (0.71–1.69) | 0.718 |
| Tight/Very tight/Poor | 1.61 (1.07–2.45) | **0.023** | 1.34 (0.80–2.27) | 0.262 |
| **Educational attainment** | | | | |
| High school or less | Reference | | Reference | |
| Some college/university | 0.74 (0.44–1.22) | 0.247 | 1.07 (0.58–1.94) | 0.826 |
| Completed college/university | 1.02 (0.75–1.39) | 0.923 | 0.89 (0.61–1.31) | 0.551 |
| Post-graduate/professional degree | 0.74 (0.47–1.15) | 0.182 | 0.96 (0.55–1.66) | 0.883 |
| **Primary language spoken** | | | | |
| English | Reference | | Reference | |
| Other | 1.41 (0.97–2.03) | 0.067 | 1.53 (0.97–2.38) | 0.063 |
| **Self-perceived health** | | | | |
| Poor | Reference | | Reference | |
| Fair | 1.04 (0.49–2.30) | 0.918 | 1.27 (0.67–2.49) | 0.481 |
| Good | 1.01 (0.50–2.15) | 0.981 | 1.17 (0.63–2.27) | 0.626 |
| Very good | 0.74 (0.37–1.59) | 0.422 | 1.00 (0.51–2.01) | 0.989 |
| Excellent | 0.49 (0.23–1.09) | 0.068 | 0.90 (0.38–2.11) | 0.801 |
| **ADG Score[4]** | | | | |
| < 3 | Reference | | Reference | |
| 3–4 | 1.20 (0.83–1.77) | 0.341 | 0.58 (0.20–1.82) | 0.335 |
| 5–6 | 0.86 (0.57–1.30) | 0.464 | 0.75 (0.28–2.18) | 0.572 |
| 7–8 | 1.10 (0.71–1.72) | 0.671 | 0.62 (0.23–1.81) | 0.360 |
| ≥ 9 | 1.29 (0.80–2.09) | 0.303 | 0.97 (0.37–2.77) | 0.945 |
| **Program type** | | | | |
| Enhanced FFS[5] | Reference | | Reference | |
| Capitation | 0.81 (0.59–1.11) | 0.187 | 0.80 (0.57–1.14) | 0.215 |
| Other | 0.58 (0.18–1.53) | 0.302 | 1.12 (0.30–3.43) | 0.851 |
| **Years with provider** | | | | |
| < 4 | Reference | | Reference | |
| 4–9 | 0.72 (0.51–1.01) | 0.056 | 0.73 (0.45–1.16) | 0.183 |
| 10–19 | 0.68 (0.48–0.96) | **0.029** | 1.09 (0.70–1.72) | 0.693 |
| ≥ 20 | 0.70 (0.50–0.98) | **0.037** | 0.67 (0.43–1.05) | 0.080 |
| **RIO category[6]** | | | | |
| Large urban | Reference | | Reference | |
| Urban | 0.94 (0.69–1.27) | 0.685 | 0.93 (0.63–1.38) | 0.733 |

*(Continued)*

**Table 4.** (Continued)

| Factors | Poor Patient Experience[1] (As of May 2019—February 2020) OR (95% CI) | | | |
|---|---|---|---|---|
| | ≤ 3 encounters, N = 1,647 | p | > 3 encounters, N = 1,038 | p |
| Small urban | 1.09 (0.75–1.59) | 0.657 | 1.08 (0.64–1.79) | 0.781 |
| Rural | 1.21 (0.72–2.03) | 0.461 | 0.79 (0.35–1.68) | 0.543 |
| **Dependency[7]** | | | | |
| 1st quintile *(least marginalized)* | Reference | | Reference | |
| 2nd quintile | 1.17 (0.81–1.70) | 0.410 | 1.28 (0.76–2.18) | 0.354 |
| 3rd quintile | 0.76 (0.51–1.13) | 0.181 | 2.44 (1.43–4.21) | **0.001** |
| 4th quintile | 0.94 (0.62–1.43) | 0.783 | 1.68 (0.96–2.95) | 0.069 |
| 5th quintile *(most marginalized)* | 0.78 (0.51–1.21) | 0.269 | 1.42 (0.80–2.52) | 0.233 |
| **Material Deprivation[8]** | | | | |
| 1st quintile *(least marginalized)* | Reference | | Reference | |
| 2nd quintile | 0.87 (0.62–1.24) | 0.447 | 0.82 (0.51–1.32) | 0.422 |
| 3rd quintile | 0.99 (0.68–1.42) | 0.949 | 0.59 (0.35–0.99) | **0.047** |
| 4th quintile | 1.03 (0.68–1.54) | 0.903 | 0.74 (0.44–1.24) | 0.255 |
| 5th quintile *(most marginalized)* | 0.72 (0.45–1.14) | 0.160 | 0.68 (0.38–1.19) | 0.180 |
| **Ethnic Concentration[9]** | | | | |
| 1st quintile *(least marginalized)* | Reference | | Reference | |
| 2nd quintile | 1.33 (0.89–1.99) | 0.158 | 0.81 (0.46–1.42) | 0.466 |
| 3rd quintile | 1.22 (0.78–1.90) | 0.384 | 0.68 (0.37–1.23) | 0.198 |
| 4th quintile | 1.50 (0.93–2.44) | 0.095 | 1.16 (0.62–2.17) | 0.650 |
| 5th quintile *(most marginalized)* | 1.68 (1.00–2.81) | **0.049** | 1.00 (0.51–1.98) | 0.996 |
| **Residential Instability[10]** | | | | |
| 1st quintile *(least marginalized)* | Reference | | Reference | |
| 2nd quintile | 1.22 (0.83–1.79) | 0.307 | 1.09 (0.65–1.82) | 0.754 |
| 3rd quintile | 1.11 (0.74–1.65) | 0.621 | 0.85 (0.49–1.47) | 0.569 |
| 4th quintile | 1.37 (0.90–2.07) | 0.143 | 1.09 (0.64–1.87) | 0.757 |
| 5th quintile *(most marginalized)* | 1.18 (0.77–1.80) | 0.440 | 1.11 (0.65–1.90) | 0.694 |
| **C-Statistic** | 0.61 | | 0.66 | |

[1]Defined as summed first quintile of patient experience scores (≤ 21)

[2]Includes telephone, email, video, electronic messaging, or other virtual means of communication with primary care provider 12 months before the survey

[3]Includes digital medical record access 12 months before the survey, including systems specific to health conditions or comprehensive records

[4]John Hopkins Aggregated Diagnosis Groups

[5]Enhanced fee for service

[6]Rurality Index of Ontario

[7]Defined by the Ontario Marginalization Index as the proportion of the population without an income generated from employment, including ratios of seniors and unemployment within the population

[8]Defined by the Ontario Marginalization Index as a measure of poverty and access to basic human needs

[9]Defined by the Ontario Marginalization Index as the concentration of recent immigrants and visible minorities

[10]Defined by the Ontario Marginalization Index as the level of family and housing instability

mechanisms to support their adoption. Despite no observed association in the summed score, telehealth access was consistent with experimental pilot studies, where patients who accessed telehealth reported similar or improved experiences and patient-provider communication [24, 39, 40]. Our findings aligned with international population-based studies and literature reviews reporting poorer experience scores among younger, financially, or ethnically marginalized adults [33, 36]. While we did not observe sex differences in patient experience, similar studies indicate minimal association [33]. Our findings were consistent with past studies

observing longstanding patient-provider relationships to play a crucial role in shared decision-making and patient experience, where longer relationships create greater trust between patients and providers [37].

Unlike our study, past studies have observed that participants in worse health have poorer experiences and fewer participatory visits [33, 36, 37]. We measured this association through self-reported health status and ADG values. However, we did not observe a noticeable effect in experience or their involvement in decision-making. Unlike our findings, a past multicenter US study has observed that adults with greater educational attainment were more likely to report shared decision-making with their providers [37]. However, the study did not account for language barriers, which played a prominent role in communication and experience with primary care providers.

The proportion of Ontario adults who had a virtual visit dramatically increased to 29.2% in 2021, compared to 1.3% in 2019 [41]. While our findings serve as a necessary baseline of experience, access to new digital health and primary care experience data allows us to expand our research beyond the pandemic. System-wide adoption of digital health may introduce health equity and digital literacy concerns. Older adults lacking computer skills or with severe illness and individuals with poor socioeconomic status, cognitive impairments, or disabilities related to language and understanding face additional barriers to accessing technology and digital health [42–45]. In response to the COVID-19 pandemic and Ontario's Digital Health Playbook, primary care digitization must be coupled with evidence and re-evaluated for its impact on patient experience while considering the socioeconomic and digital literacy barriers to patient-centred care and digital health adoption [46].

## Limitations

The study uses data from before the COVID-19 pandemic, therefore serving as a baseline for future research examining post-pandemic trends in experience and digital health access. Most study participants were older, while 16.7% of Ontarians are ≥65 years of age [47]. The older demographic may limit the adoption of digital health and bias the sample toward adults who frequently access primary care and have longstanding patient-provider relationships [48, 49]. The survey is provided to English or French-speaking Ontarians in private residences with a valid address in RPDB, 16 years or older, with a valid health insurance card and active phone number [31]. The HCES sampling frame would potentially exclude individuals who do not speak English or French, institutionalized individuals, refugees, persons experiencing homelessness, some Indigenous persons, and persons without access to a phone [28, 31]. Further, ISR did not survey individuals who were not physically or mentally healthy enough, resulting in an underrepresentation of persons with cognitive impairments and other conditions [28].

Patient experience encompasses all interactions with primary care and is influenced by expectations of care [8]. We were unable to measure associations longitudinally or through provider characteristics. Future research should observe long-term associations with experience and implement further health provider and practice factors suggested by frameworks, including the 4 Ps of Patient Experience [50]. Our analysis used complete cases of data without any further imputation. The full and analyzed cohorts were generally consistent in demographic characteristics (Table 2).

## Supporting information

**S1 Appendix. Summary of coding done on dependent and independent variables.**
(DOCX)

**S2 Appendix. Histogram of patient experience scores with binary groupings.**
(DOCX)

**S3 Appendix. Odds ratios and confidence intervals for single-item experience models.**
(DOCX)

**S4 Appendix. Strobe statement.**
(DOCX)

**S5 Appendix. Analytical code.**
(DOCX)

**S6 Appendix. Dataset creation plan.**
(DOCX)

## Acknowledgments

This study was supported by ICES, which is funded by an annual grant from the Ontario Ministry of Health (MOH) and the Ministry of Long-Term Care (MLTC). Parts of this material are based on data and information compiled and provided by the Canadian Institute for Health Information (CIHI) and the Ontario Ministry of Health. The analyses, conclusions, opinions, and statements expressed herein are solely those of the author and do not reflect those of the funding or data sources; no endorsement is intended or should be inferred.

The Johns Hopkins ACG® System Version 10 was used to provide Aggregated Diagnosis Groups (ADGs).

We thank the Toronto Community Health Profiles Partnership for providing access to the Ontario Marginalization Index.

## Author Contributions

**Conceptualization:** Zain Pasat, Andrew P. Costa.

**Data curation:** Anastasia Gayowsky.

**Formal analysis:** Zain Pasat.

**Funding acquisition:** Andrew P. Costa.

**Investigation:** Zain Pasat.

**Methodology:** Zain Pasat, Andrew P. Costa.

**Supervision:** Cynthia Lokker, Jean-Eric Tarride, Andrew P. Costa.

**Visualization:** Zain Pasat.

**Writing – original draft:** Zain Pasat.

**Writing – review & editing:** Zain Pasat, Chi-Ling Joanna Sinn, Bahram Rahman, Anastasia Gayowsky, Cynthia Lokker, Jean-Eric Tarride, Mohamed Alarakhia, Andrew P. Costa.

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
