## [Decision Letter · Decision Letter 0]

28 Nov 2023

PONE-D-23-26136The relationship between patient experience and real-world digital health access in primary care: A population-based cross-sectional studyPLOS ONE

Dear Dr. Pasat,

Thank you for submitting your manuscript to PLOS ONE. After careful consideration, we feel that it has merit but does not fully meet PLOS ONE’s publication criteria as it currently stands. Therefore, we invite you to submit a revised version of the manuscript that addresses the points raised during the review process.

In the light of comments from two of the peer reviewers, your manuscript needs to address some important points as well as bettering of language expression.

We look forward to receiving your revised manuscript.

Kind regards,

Muhammad Farooq Umer, PhD Epidemiology and Health Statistics

Academic Editor

PLOS ONE

3. "Please provide additional details regarding participant consent. In the ethics statement in the Methods and online submission information, please ensure that you have specified (1) whether consent was informed and (2) what type you obtained (for instance, written or verbal, and if verbal, how it was documented and witnessed). If your study included minors, state whether you obtained consent from parents or guardians. If the need for consent was waived by the ethics committee, please include this information.

“This study was supported by ICES, which is funded by an annual grant from the Ontario Ministry of Health (MOH) and the Ministry of Long-Term Care (MLTC). Parts of this material are based on data and information compiled and provided by the Canadian Institute for Health Information (CIHI) and the Ontario Ministry of Health. The analyses, conclusions, opinions, and statements expressed herein are solely those of the author and do not reflect those of the funding or data sources; no endorsement is intended or should be inferred.

The Johns Hopkins ACG® System Version 10 was used to provide Aggregated Diagnosis Groups (ADGs).

We thank the Toronto Community Health Profiles Partnership for providing access to the Ontario Marginalization Index.

This work was supported by the St. Joseph’s Health System Centre for Integrated Care (https://stjoescic.ca/) that is funded by St. Joseph’s Health System (https://sjhs.ca/), and the Schlegel Chair in Clinical Epidemiology and Aging, McMaster University.”

Reviewers' comments:

Reviewer's Responses to Questions

**Comments to the Author**

1. Is the manuscript technically sound, and do the data support the conclusions?

Reviewer #1: Yes

Reviewer #2: Yes

2. Has the statistical analysis been performed appropriately and rigorously? 

Reviewer #1: Yes

Reviewer #2: N/A

3. Have the authors made all data underlying the findings in their manuscript fully available?

Reviewer #1: No

Reviewer #2: Yes

4. Is the manuscript presented in an intelligible fashion and written in standard English?

Reviewer #1: Yes

Reviewer #2: No

5. Review Comments to the Author

Reviewer #1: This is an interesting study, taking advantage of pre-pandemic survey information and tying it to actual health administrative data, which is quite innovative.

The paper is written clearly and I think it makes an important contribution even though there were not many strong associations observed. I was not aware of the HCES survey prior to reading this paper. It is important to recognize, as authors do, how few people were using the electronic booking and records viewing prior to the pandemic. It is tricky to interpret differences or lack of them for these outcomes given how infrequent they were, but important to document how few people in this phone survey reported them. Digital communication through telehealth, however is important to understand pre-pandemic and there was a substantial subgroup who reported this occurring.

My only comments in revising this manuscript would be:

limitation that the phone survey also may have excluded people with certain disabilities unless efforts were made to make the phone interview accessible across disability groups. This would include some people with cognitive disabilities and autistic people who have difficulties with phone based interviews (and phone based medical appointments).

The survey was conducted with people who had residential phone numbers so this would likely, especially in later years, exclude people who only pay for a cell phone which would be younger people, as observed, and also people with lower income who pay only for a cell phone.

There was a finding that younger people had more difficulties. It may be worth commenting that as the other aspects of digital care (booking, record reviewing and video telehealth) became more prominent, this would be a barrier to older and not younger people. They study looked at telephone based care, and younger people are less comfortable with phone but more comfortable with other ways to communicate digitally (e.g., text). In a similar vein, digital care can pose problems to certain disability groups, while also being more accessible to other disability groups. In referring to relevant groups from the international literature, it would also be important to reference what is known about digital healthcare and people with disabilities.

Reviewer #2: The manuscript seems to be an excerpt of a thesis. It is well written and contains good information. Recent references can be added. The language needs to be reviewed and corrected as per the journal's guidelines.

6. PLOS authors have the option to publish the peer review history of their article (what does this mean?). If published, this will include your full peer review and any attached files.

Reviewer #1: No

Reviewer #2: **Yes: **Muhammad Faheemuddin

---

## [Author Response · Author response to Decision Letter 0]

22 Jan 2024

Thank you for your feedback on our research article. As requested in the decision letter, our responses to the editor's and reviewers' comments are in the included file "Response to Reviewers." We have revised the manuscript accordingly to address your concerns and feedback.

---

## [Decision Letter · Decision Letter 1]

5 Feb 2024

The relationship between patient experience and real-world digital health access in primary care: A population-based cross-sectional study

PONE-D-23-26136R1

Dear Dr. Pasat,

We’re pleased to inform you that your manuscript has been judged scientifically suitable for publication and will be formally accepted for publication once it meets all outstanding technical requirements.

Kind regards,

Muhammad Farooq Umer, PhD Epidemiology and Health Statistics

Academic Editor

PLOS ONE

Additional Editor Comments (optional):

Reviewers' comments:

Reviewer's Responses to Questions

**Comments to the Author**

1. If the authors have adequately addressed your comments raised in a previous round of review and you feel that this manuscript is now acceptable for publication, you may indicate that here to bypass the “Comments to the Author” section, enter your conflict of interest statement in the “Confidential to Editor” section, and submit your "Accept" recommendation.

Reviewer #1: All comments have been addressed

2. Is the manuscript technically sound, and do the data support the conclusions?

Reviewer #1: (No Response)

3. Has the statistical analysis been performed appropriately and rigorously? 

Reviewer #1: (No Response)

4. Have the authors made all data underlying the findings in their manuscript fully available?

Reviewer #1: (No Response)

5. Is the manuscript presented in an intelligible fashion and written in standard English?

Reviewer #1: (No Response)

6. Review Comments to the Author

Reviewer #1: (No Response)

7. PLOS authors have the option to publish the peer review history of their article (what does this mean?). If published, this will include your full peer review and any attached files.

Reviewer #1: No

---

## [Editor Report · Acceptance letter]

25 Apr 2024

PONE-D-23-26136R1 

PLOS ONE

Dear Dr. Pasat, 

I'm pleased to inform you that your manuscript has been deemed suitable for publication in PLOS ONE. Congratulations! Your manuscript is now being handed over to our production team.

Kind regards, 

on behalf of

Dr. Muhammad Farooq Umer 

Academic Editor

PLOS ONE